# Some Properties of Interior and Closure in General Topology

**Soon-Mo Jung** [1],*,†  **and Doyun Nam** [2],†

[1]   Mathematics Section, College of Science and Technology, Hongik University, Sejong 30016, Korea
[2]   Department of Mathematical Sciences, Seoul National University, Seoul 08826, Korea
*    Correspondence: smjung@hongik.ac.kr; Tel.: +82-44-860-2584
†    These authors contributed equally to this work.

**Abstract:** We present the necessary and sufficient conditions that the intersection of an open set and a closed set becomes either an open set or a closed set. As their dualities, we further introduce the necessary and sufficient conditions that the union of a closed set and an open set becomes either a closed set or an open set. Moreover, we give some necessary and sufficient conditions for the validity of $U^\circ \cup V^\circ = (U \cup V)^\circ$ and $\overline{U} \cap \overline{V} = \overline{U \cap V}$. Finally, we introduce a necessary and sufficient condition for an open subset of a closed subspace of a topological space to be open. As its duality, we also give a necessary and sufficient condition for a closed subset of an open subspace to be closed.

**Keywords:** open set; closed set; duality; union; intersection; topological space

## 1. Introduction

This present paper has been written based on the first author's 2016 paper [1], and the present paper has been completed with many enhancements and extensions of the previous paper [1]. In particular, the sufficient conditions of the previous paper have been changed to necessary and sufficient conditions in this paper, which greatly improved the completeness of this paper.

In Section 2 of the present paper, we introduce some necessary and sufficient conditions that the intersection of an open set and a closed set of a topological space becomes either an open set or a closed set, even though it seems to be a typically classical subject. Symmetrically, we also present some necessary and sufficient conditions that the union of a closed set and an open set becomes either a closed set or an open set.

The following are relatively well known:

$$U^\circ \cap V^\circ = (U \cap V)^\circ,$$
$$U^\circ \cup V^\circ \subset (U \cup V)^\circ, \tag{1}$$
$$\overline{U} \cup \overline{V} = \overline{U \cup V},$$
$$\overline{U} \cap \overline{V} \supset \overline{U \cap V} \tag{2}$$

for subsets $U$ and $V$ of $X$ (refer to [2–4]).

The inclusion (1) or (2) holds true for 'strict inclusion' when $U$ and $V$ are some particular subsets of a topological space $X$. For instance, we may set $U = \mathbb{Q}$, $V = \mathbb{R}\backslash\mathbb{Q}$, and $X = \mathbb{R}$. So $U^\circ = \mathbb{Q}^\circ = \varnothing$, $V^\circ = (\mathbb{R}\backslash\mathbb{Q})^\circ = \varnothing$, and $(U \cup V)^\circ = \mathbb{R}^\circ = \mathbb{R}$. Thus, it holds that $U^\circ \cup V^\circ \neq (U \cup V)^\circ$. Similarly, it follows that $\overline{U} = \overline{\mathbb{Q}} = \mathbb{R}$, $\overline{V} = \overline{\mathbb{R}\backslash\mathbb{Q}} = \mathbb{R}$, and $\overline{U \cap V} = \overline{\mathbb{Q} \cap (\mathbb{R}\backslash\mathbb{Q})} = \overline{\varnothing} = \varnothing$. So, it holds that $\overline{U} \cap \overline{V} \neq \overline{U \cap V}$. These examples show that, in general, the equality symbol may not be true in (1)

and (2). However, in many practical applications, it would be important for us to know the conditions that we can expect equality symbols in (1) and (2). This is the motivation of Sections 3 and 4 of the present paper. Indeed in Sections 3 and 4, we will find the necessary and sufficient conditions for them.

What is the condition that an open subset of a closed set becomes an open set? Or what is the condition that a closed subset of an open set becomes a closed set? Partial answers to these questions can be found in Section 5. In Section 5, we introduce a necessary and sufficient condition for which an open subset of a closed subspace of a topological space be open. As its duality, we also introduce a necessary and sufficient condition for a closed subset of an open subspace of a topological space to be closed.

If there is no other specification in the present paper, suppose $X$ is a topological space. For each subset $U$ of a topological space $X$, $U^\circ$ and $\overline{U}$ denote the interior and the closure of $U$ in $X$, respectively.

## 2. Preliminaries

The union (or intersection) of finitely many open subsets is open. As its duality, the intersection (or union) of finitely many closed subsets is closed. Now what can we say about the union (or intersection) of an open subset and a closed subset of a topological space?

In the following theorem, roughly speaking, we prove that the intersection of a connected open set and a closed set is open if and only if the closed set includes the open set.

**Theorem 1.** *Assume that $F$ is a closed subset and $G$ is a nonempty connected open subset of a topological space $X$. Then, $F \cap G$ is a nonempty open subset of $X$ if and only if $G \subset F$.*

**Proof.** Suppose $F \cap G$ is a nonempty open subset of $X$ but $G \not\subset F$, that is, $G \backslash F = G \cap X \backslash F$ is a nonempty open subset of $X$. Then the relation $G = (G \backslash F) \cup (F \cap G)$ would lead to the contradiction that $G$ is the union of two nonempty disjoint open sets, that is, $G$ is not connected. Thus if $F \cap G$ is a nonempty open subset of $X$, then $G \subset F$. The converse is obvious. $\square$

We may regard the following corollary as the duality of Theorem 1.

**Corollary 1.** *Assume that $F$ is a proper closed subset, $G$ is an open subset of a topological space $X$, and that $X \backslash F$ is connected. Then, $F \cup G$ is a proper closed subset of $X$ if and only if $G \subset F$.*

**Proof.** We know that $X \backslash F$ is a nonempty connected open subset and $X \backslash G$ is a closed subset of $X$ from the hypotheses. Thus, by substituting $X \backslash F$ for $G$ and $X \backslash G$ for $F$ in Theorem 1, we obtain a true statement: $X \backslash F \cap X \backslash G$ is nonempty and open in $X$ if and only if $X \backslash F \subset X \backslash G$. From this statement, we can easily derive the desired result. $\square$

The next theorem is another version of Theorem 1. In other words, we prove that the intersection of a closed set and an open set is closed if and only if the open set includes the closed one.

**Theorem 2.** *Assume that $F$ is a nonempty connected closed subset and $G$ is an open subset of a topological space $X$. Then, $F \cap G$ is a nonempty closed subset of $X$ if and only if $F \subset G$.*

**Proof.** Assume that $F \cap G$ is a nonempty closed subset of $X$ but $F \not\subset G$, that is, $F \backslash G = F \cap X \backslash G$ is a nonempty closed subset of $X$. Then the relation $F = (F \backslash G) \cup (F \cap G)$ would lead to the contradiction that $F$ is the union of two nonempty disjoint closed sets, that is, $F$ is not connected. Thus if $F \cap G$ is nonempty and closed in $X$, then $F \subset G$. The converse is obvious. $\square$

We may regard the following corollary as the duality of Theorem 2. More precisely, in the following corollary we deal with the openness of the union of an open subset and a closed subset of a topological space, which is another version of Corollary 1.

**Corollary 2.** *Assume that F is a closed subset, G is an open proper subset of a topological space X, and that* $X \backslash G$ *is connected. Then,* $F \cup G$ *is a proper open subset of X if and only if* $F \subset G$.

**Proof.** We know that $X \backslash F$ is an open subset and $X \backslash G$ is a nonempty connected closed subset of $X$ from our assumptions. Thus, by substituting $X \backslash F$ for $G$ and $X \backslash G$ for $F$ in Theorem 2, we obtain a true statement: $X \backslash F \cap X \backslash G$ is nonempty and closed in $X$ if and only if $X \backslash G \subset X \backslash F$. From this statement, we can easily derive the desired result. □

The next theorem is not entirely new, but we introduce the proof of this theorem to improve the completeness of this paper. This theorem is essential to prove Theorem 5. We remember that $X$ is a topological space if there is no other special description.

**Theorem 3.** *Assume that U is a subset of X. Then*

(*a*)　$X \backslash U^\circ = \overline{X \backslash U}$;
(*b*)　$X \backslash \overline{U} = (X \backslash U)^\circ$.

**Proof.** (*a*) We can easily prove this assertion by using the following equation:

$$X \backslash U^\circ = X \backslash \bigcup_{\substack{V \text{ is open} \\ V \subset U}} V = \bigcap_{\substack{V \text{ is open} \\ V \subset U}} X \backslash V = \bigcap_{\substack{W \text{ is closed} \\ X \backslash U \subset W}} W = \overline{X \backslash U}$$

(*b*) We can also prove the second argument using the following equation:

$$X \backslash \overline{U} = X \backslash \bigcap_{\substack{W \text{ is closed} \\ U \subset W}} W = \bigcup_{\substack{W \text{ is closed} \\ U \subset W}} X \backslash W = \bigcup_{\substack{V \text{ is open} \\ V \subset X \backslash U}} V = (X \backslash U)^\circ$$

which completes the proof of this theorem. □

The following lemma is often used in Section 3.

**Lemma 1.** *If A, B, and C are arbitrary subsets of a topological space, then the following equalities hold:*

(*a*)　$(A \backslash B) \backslash C = A \backslash (B \cup C)$;
(*b*)　$A \backslash (B \backslash C) = (A \backslash B) \cup (A \cap C)$;
(*c*)　$(A \cup B) \backslash C = A \backslash C \cup B \backslash C$ *and* $(A \cap B) \backslash C = A \backslash C \cap B \backslash C$;
(*d*)　$A \backslash (B \cap C) = A \backslash B \cup A \backslash C$ *and* $A \backslash (B \cup C) = A \backslash B \cap A \backslash C$.

**Proof.** Equalities (*c*) and (*d*) are easy to prove, thus we omit their proofs. The equality (*a*) follows from the equation

$$(A \backslash B) \backslash C = (A \cap X \backslash B) \cap X \backslash C = A \cap (X \backslash B \cap X \backslash C)$$
$$= A \cap (X \backslash (B \cup C)) = A \backslash (B \cup C).$$

The equality (*b*) comes from the equation

$$A \backslash (B \backslash C) = A \backslash (B \cap X \backslash C) = A \cap (X \backslash (B \cap X \backslash C))$$
$$= A \cap (X \backslash B \cup C) = (A \cap X \backslash B) \cup (A \cap C)$$
$$= (A \backslash B) \cup (A \cap C),$$

and we finish the proof. □

## 3. Interiors

In this section, let $X$ be a topological space if there is no other special description. We know that $U^\circ \cup V^\circ \subset (U \cup V)^\circ$ is true for all subsets $U$ and $V$ of $X$ in the normal case. In the next theorem, we deal with some necessary and sufficient conditions that allow the union of interiors of two subsets to equal the interior of union of those two subsets. Although it is not clear at this point in what areas this equality can be used, this equality is very interesting from a theoretical point of view.

We may show that each 'convex' subset of $\mathbb{R}^3$ satisfies the condition $(b)$ or $(c)$ of the following theorem.

**Theorem 4.** *If $U$ and $V$ are arbitrary subsets of $X$, then statements $(a)$, $(b)$, and $(c)$ are equivalent:*

$(a)$　$U^\circ \cup V^\circ = (U \cup V)^\circ$;
$(b)$　$(\partial U \backslash V^\circ) \cup (\partial V \backslash U^\circ) = \partial(U \cup V)$;
$(c)$　$\partial U \backslash V^\circ = \overline{U} \backslash (U \cup V)^\circ$ *and* $\partial V \backslash U^\circ = \overline{V} \backslash (U \cup V)^\circ$.

**Proof.** We will prove in turn $(c) \Rightarrow (b)$, $(b) \Rightarrow (a)$, and $(a) \Rightarrow (c)$. If $(c)$ holds, it then follows from Lemma 1 $(c)$ that

$$(\partial U \backslash V^\circ) \cup (\partial V \backslash U^\circ) = (\overline{U} \backslash (U \cup V)^\circ) \cup (\overline{V} \backslash (U \cup V)^\circ) = (\overline{U} \cup \overline{V}) \backslash (U \cup V)^\circ$$
$$= (\overline{U \cup V}) \backslash (U \cup V)^\circ = \partial(U \cup V),$$

which proves $(b)$. On account of Lemma 1 $(a)$ and $(c)$, we know that

$$\partial U \backslash V^\circ = (\overline{U} \backslash U^\circ) \backslash V^\circ = \overline{U} \backslash (U^\circ \cup V^\circ),$$
$$\partial V \backslash U^\circ = (\overline{V} \backslash V^\circ) \backslash U^\circ = \overline{V} \backslash (U^\circ \cup V^\circ), \tag{3}$$

and hence

$$(\partial U \backslash V^\circ) \cup (\partial V \backslash U^\circ) = \overline{U} \backslash (U^\circ \cup V^\circ) \cup \overline{V} \backslash (U^\circ \cup V^\circ)$$
$$= (\overline{U} \cup \overline{V}) \backslash (U^\circ \cup V^\circ) \tag{4}$$
$$= \overline{U \cup V} \backslash (U^\circ \cup V^\circ).$$

If $(b)$ is true, then we know that $\partial(U \cup V) = \overline{U \cup V} \backslash (U^\circ \cup V^\circ)$ from $(b)$ and $(4)$. Thus

$$(U \cup V)^\circ = \overline{U \cup V} \backslash \partial(U \cup V) = \overline{U \cup V} \backslash (\overline{U \cup V} \backslash (U^\circ \cup V^\circ)) = U^\circ \cup V^\circ,$$

which proves $(a)$. If $(a)$ holds, then by $(3)$ we can easily check that $(c)$ holds. □

We are now ready to prove a variant of Theorem 3 for subsets of topological space. In fact, we will prove $U^\circ \backslash \overline{V} = (U \backslash V)^\circ$ and $\overline{U} \backslash V^\circ = \overline{U \backslash V}$ under certain conditions.

**Theorem 5.** *If $U$ and $V$ are arbitrary subsets of $X$, then $U^\circ \backslash \overline{V} = (U \backslash V)^\circ$. Furthermore, the statements $(a)$, $(b)$, $(c)$, and $(d)$ are equivalent:*

$(a)$　$\overline{U} \backslash V^\circ = \overline{U \backslash V}$;
$(b)$　$(X \backslash U)^\circ \cup V^\circ = ((X \backslash U) \cup V)^\circ$;
$(c)$　$\overline{U} \cap \overline{X \backslash V} = \overline{U \cap (X \backslash V)}$;
$(d)$　$\partial U \backslash V^\circ = \overline{X \backslash U} \backslash ((X \backslash U) \cup V)^\circ$ *and* $\overline{U} \cap \partial V = \overline{V} \backslash ((X \backslash U) \cup V)^\circ$.

**Proof.** By Theorem 3 $(b)$, we obtain

$$U^\circ \backslash \overline{V} = U^\circ \cap (X \backslash \overline{V}) = U^\circ \cap (X \backslash V)^\circ = (U \cap X \backslash V)^\circ = (U \backslash V)^\circ.$$

And we shall prove $(a) \Rightarrow (b) \Rightarrow (c) \Rightarrow (a)$, and then $(b) \Leftrightarrow (d)$. If $(a)$ holds, then

$$(X \backslash U)^\circ \cup V^\circ = (X \backslash \overline{U}) \cup V^\circ = X \backslash (\overline{U} \cap (X \backslash V^\circ)) = X \backslash (\overline{U} \backslash V^\circ)$$
$$= X \backslash \overline{U \backslash V} = X \backslash \overline{U \cap X \backslash V} = (X \backslash (U \cap X \backslash V))^\circ$$
$$= ((X \backslash U) \cup V)^\circ,$$

which proves $(b)$. If $(b)$ is true, then

$$\overline{U} \cap \overline{X \backslash V} = \overline{X \backslash (X \backslash U)} \cap \overline{X \backslash V} = X \backslash (X \backslash U)^\circ \cap X \backslash V^\circ = X \backslash ((X \backslash U)^\circ \cup V^\circ)$$
$$= X \backslash ((X \backslash U) \cup V)^\circ = \overline{X \backslash ((X \backslash U) \cup V)} = \overline{(X \backslash X \backslash U) \cap X \backslash V}$$
$$= \overline{U \cap (X \backslash V)},$$

which proves $(c)$. If $(c)$ holds, then

$$\overline{U} \backslash V^\circ = \overline{U} \cap X \backslash V^\circ = \overline{U} \cap \overline{X \backslash V} = \overline{U \cap (X \backslash V)} = \overline{U \backslash V},$$

which proves $(a)$. Hence, we proved that $(a) \Leftrightarrow (b) \Leftrightarrow (c)$.

By Theorem 3 $(b)$, we know

$$\overline{U} \cap \partial V = \partial V \cap (X \backslash X \backslash \overline{U}) = \partial V \cap (X \backslash (X \backslash U)^\circ) = \partial V \backslash (X \backslash U)^\circ.$$

And it is obvious that $\partial U = \partial (X \backslash U)$. Therefore $(d)$ is equivalent to

$$\partial (X \backslash U) \backslash V^\circ = \overline{X \backslash U} \backslash ((X \backslash U) \cup V)^\circ \quad \text{and} \quad \partial V \backslash (X \backslash U)^\circ = \overline{V} \backslash ((X \backslash U) \cup V)^\circ.$$

By substituting $X \backslash U$ for $U$ in the Theorem 4 $(a)$ and $(c)$, we conclude that the above equality is equivalent to $(b)$. Hence, we proved $(b) \Leftrightarrow (d)$. $\square$

## 4. Closures

It is well and widely known that $\overline{U} \cap \overline{V} \supset \overline{U \cap V}$ for all subsets $U$ and $V$ of $X$. In the following theorem, we examine some necessary and sufficient conditions that allow the intersection of closures of two subsets to be equal to the closure of intersection of those two subsets. In short, the following theorem is the counterpart of Theorem 4 for the relation $\overline{U} \cap \overline{V} = \overline{U \cap V}$.

We can show that every 'convex' subset of $\mathbb{R}^3$ satisfies the condition $(b)$ or $(c)$ of the following theorem. Indeed, using the duality property, we can apply the similar idea of Theorem 4 to prove the following theorem.

**Theorem 6.** *If $U$ and $V$ are arbitrary subsets of $X$, then statements $(a)$, $(b)$, and $(c)$ are equivalent:*

$(a)$  $\overline{U} \cap \overline{V} = \overline{U \cap V}$;
$(b)$  $\partial (U \cap V) = (\partial U \cup \partial V) \cap (\overline{U} \cap \overline{V})$;
$(c)$  $\overline{U} \cap \partial V = \overline{U \cap V} \backslash V^\circ$ *and*  $\overline{V} \cap \partial U = \overline{U \cap V} \backslash U^\circ$.

**Proof.** We can check that

$$(\overline{U} \cap \overline{V}) \backslash (U \cap V)^\circ = (\overline{U} \cap \overline{V}) \backslash (U^\circ \cap V^\circ)$$
$$= (\overline{U} \cap \overline{V}) \backslash V^\circ \cup (\overline{U} \cap \overline{V}) \backslash U^\circ$$
$$= (\overline{U} \cap \overline{V} \cap X \backslash V^\circ) \cup (\overline{U} \cap \overline{V} \cap X \backslash U^\circ) \qquad (5)$$
$$= (\overline{U} \cap \partial V) \cup (\overline{V} \cap \partial U).$$

Also, we can check

$$(\partial U \cup \partial V) \cap (\overline{U} \cap \overline{V}) = (\partial V \cap \overline{U} \cap \overline{V}) \cup (\partial U \cap \overline{U} \cap \overline{V})$$
$$= (\overline{U} \cap \partial V) \cup (\overline{V} \cap \partial U). \tag{6}$$

We will prove in turn $(c) \Rightarrow (b)$, $(b) \Rightarrow (a)$, and $(a) \Rightarrow (c)$. If $(c)$ holds, then it follows from (6) and $(c)$ that

$$(\partial U \cup \partial V) \cap (\overline{U} \cap \overline{V}) = (\overline{U} \cap \partial V) \cup (\overline{V} \cap \partial U) = (\overline{U \cap V} \backslash V^\circ) \cup (\overline{U \cap V} \backslash U^\circ)$$
$$= \overline{U \cap V} \backslash (U^\circ \cap V^\circ) = \overline{U \cap V} \backslash (U \cap V)^\circ = \partial(U \cap V),$$

which proves $(b)$. If $(b)$ is true, then by combining $(b)$, (5), and (6), we get the equality

$$\partial(U \cap V) = (\overline{U} \cap \overline{V}) \backslash (U \cap V)^\circ.$$

We know that $(U \cap V)^\circ \subset U \cap V \subset \overline{U} \cap \overline{V}$. Hence, we conclude that

$$\overline{U \cap V} = (U \cap V)^\circ \cup \partial(U \cap V) = (U \cap V)^\circ \cup ((\overline{U} \cap \overline{V}) \backslash (U \cap V)^\circ) = \overline{U} \cap \overline{V},$$

which proves $(a)$. If $(a)$ holds, then

$$\overline{U \cap V} \backslash V^\circ = (\overline{U} \cap \overline{V}) \backslash V^\circ = \overline{U} \cap \overline{V} \cap X \backslash V^\circ = \overline{U} \cap \partial V,$$

and similarly we prove that $\overline{U \cap V} \backslash U^\circ = \overline{V} \cap \partial U$, which proves $(c)$. $\square$

## 5. Openness of Subset of Subspace

Let $X$ be a topological space. In the following theorem, we introduce sufficient conditions under which the intersection of two subsets is an open set.

**Theorem 7.** *If arbitrary subsets $U$ and $V$ of $X$ satisfy the properties:*

*(a)* $((X \backslash U) \cup V)^\circ = (X \backslash U)^\circ \cup V^\circ$;
*(b)* $U$ *is open*;
*(c)* $\overline{U} \backslash V$ *is closed*,

*then $U \cap V$ is an open set.*

**Proof.** Condition $(a)$ is equivalent to $\overline{U} \backslash V^\circ = \overline{U \backslash V}$ by Theorem 5. Thus, we get the relation

$$U \backslash V \subset \overline{U} \backslash V \subset \overline{U} \backslash V^\circ = \overline{U \backslash V}.$$

Because $\overline{U} \backslash V$ is closed by condition $(c)$ and it includes $U \backslash V$, we obtain the relation

$$\overline{U \backslash V} \subset \overline{U} \backslash V.$$

Thus, it holds that $\overline{U} \backslash V = \overline{U} \backslash V^\circ = \overline{U \backslash V}$. By using Lemma 1 $(b)$ twice, we obtain

$$\overline{U} \cap V = (\overline{U} \backslash \overline{U}) \cup (\overline{U} \cap V) = \overline{U} \backslash (\overline{U} \backslash V)$$
$$= \overline{U} \backslash (\overline{U} \backslash V^\circ) = (\overline{U} \backslash \overline{U}) \cup (\overline{U} \cap V^\circ)$$
$$= \overline{U} \cap V^\circ.$$

Thus $U \cap V = U \cap (\overline{U} \cap V) = U \cap (\overline{U} \cap V^\circ) = U \cap V^\circ$, and because $U$ is open, so is $U \cap V$. $\square$

It is obvious that if $Z$ is a closed (an open) subset of $Y$ and $Y$ is a closed (an open) subspace of a topological space $X$, then $Z$ is closed (open) in $X$.

The following theorem deals with a necessary and sufficient condition that an open subset of a closed subspace of $X$ becomes an open subset of $X$.

**Theorem 8.** *Assume that $Z$ is an open subset of $Y$, where $Y$ is a closed subspace of $X$. Then, $Z$ is open in $X$ if and only if $Z \subset Y^\circ$, where $Y^\circ$ denotes the interior of $Y$ in $X$.*

**Proof.** Assume that $Z$ is open in $X$. Because $Y^\circ$ is the union of all subsets of $X$ which are open in $X$ and contained in $Y$, it follows that $Z \subset Y^\circ$. On the other hand, because $Z$ is open in $Y$, there is an open subset $U$ of $X$ satisfying $Z = U \cap Y$. If $Z \subset Y^\circ$, then

$$Z = Z \cap Y^\circ = U \cap Y \cap Y^\circ = U \cap Y^\circ.$$

Since $U$ and $Y^\circ$ is open in $X$, we conclude that $Z$ is open in $X$.  □

The next theorem provides the necessary and sufficient condition that a closed subset of the open subspace of $X$ becomes a closed subset of $X$.

**Theorem 9.** *Assume that $Z$ is a closed subset of $Y$, where $Y$ is an open subspace of $X$. Then, $Z$ is closed in $X$ if and only if $\overline{Z} \subset Y$, where $\overline{Z}$ denotes the closure of $Z$ in $X$.*

**Proof.** If $Z$ is closed in $X$, then it is obvious that $Z = \overline{Z} \subset Y$. On the other hand, because $Z$ is closed in $Y$, there exists a closed subset $V$ of $X$ satisfying $Z = V \cap Y$. If $\overline{Z} \subset Y$, then

$$Z = Z \cap \overline{Z} = V \cap Y \cap \overline{Z} = V \cap \overline{Z}.$$

Since $V$ and $\overline{Z}$ is closed in $X$, we conclude that $Z$ is closed in $X$.  □

Obviously, $U \cup V$ is an open set whenever both $U$ and $V$ are open sets. Conversely, if $U \cup V$ is open, under what conditions can we expect that both $U$ and $V$ are open?

**Theorem 10.** *Assume that $U$ and $V$ are arbitrary subsets of $X$ which are mutually separated in $X$, that is, $\overline{U} \cap V = U \cap \overline{V} = \emptyset$. If $U \cup V$ is open, then both $U$ and $V$ are open.*

**Proof.** Because $U$ and $V$ are mutually separated, it follows that $U \subset X \backslash \overline{V}$. Since $U \cup V$ and $X \backslash \overline{V}$ are open, their intersection

$$(U \cup V) \cap (X \backslash \overline{V}) = (U \cap X \backslash \overline{V}) \cup (V \cap X \backslash \overline{V}) = U \cup \emptyset = U$$

is open. In the same way, we can prove that $V$ is open.  □

## 6. Discussion

This present paper was based on the first author's 2016 paper [1] and the present paper has been completed with many enhancements and extensions of the previous paper [1]. In particular, the sufficient conditions of the previous paper have been changed to necessary and sufficient conditions in this paper, which greatly improved the completeness of this paper.

On the website [5], we found another necessary and sufficient condition

$$\partial U \cap \partial V \subset \partial(U \cup V)$$

under which the equality sign holds in the relation (1). Apparently, this condition differs from the conditions $(b)$ and $(c)$ of Theorem 4. A proof for this condition is presented in the website. However,

for the sake of completeness of this paper, we will present a simpler and shorter proof of this assertion in the following theorem.

**Theorem 11.** *If $U$ and $V$ are arbitrary subsets of $X$, then the statements $(a)$ and $(b)$ are equivalent:*

$(a)$  $U^\circ \cup V^\circ = (U \cup V)^\circ$;
$(b)$  $\partial U \cap \partial V \subset \partial(U \cup V)$.

**Proof.** Obviously, it holds that

$$\partial U \cap \partial V = (\overline{U} \cap X \backslash U^\circ) \cap (\overline{V} \cap X \backslash V^\circ) = \overline{U} \cap \overline{V} \cap X \backslash U^\circ \cap X \backslash V^\circ$$
$$= \overline{U} \cap \overline{V} \cap X \backslash (U^\circ \cup V^\circ) = (\overline{U} \cap \overline{V}) \backslash (U^\circ \cup V^\circ),$$

and $\partial(U \cup V) = \overline{U \cup V} \backslash (U \cup V)^\circ = (\overline{U} \cup \overline{V}) \backslash (U \cup V)^\circ$. Thus $(b)$ is equivalent to

$$(\overline{U} \cap \overline{V}) \backslash (U^\circ \cup V^\circ) \subset (\overline{U} \cup \overline{V}) \backslash (U \cup V)^\circ. \tag{7}$$

It is obvious that $\overline{U} \cap \overline{V} \subset \overline{U} \cup \overline{V}$. Hence if $(a)$ holds, then (7) holds.

On the other hand, assume that (7) holds. It implies that

$$\text{if } x \in (U \cup V)^\circ \text{ and } x \in \overline{U} \cap \overline{V} \text{ then } (x \in \overline{U} \cup \overline{V} \text{ and so}) \; x \in U^\circ \cup V^\circ. \tag{8}$$

We shall show that if $x \in (U \cup V)^\circ$ and $x \notin \overline{U} \cap \overline{V}$ then $x \in U^\circ \cup V^\circ$, which proves $U^\circ \cup V^\circ \supset (U \cup V)^\circ$ in cooperation with (8). Because of $U^\circ \cup V^\circ \subset (U \cup V)^\circ$, it follows that $U^\circ \cup V^\circ = (U \cup V)^\circ$. Thus, it only needs to show

$$(U \cup V)^\circ \cap (X \backslash (\overline{U} \cap \overline{V})) \subset U^\circ \cup V^\circ.$$

Indeed, we can check that

$$
\begin{aligned}
(U \cup V)^\circ \cap (X \backslash (\overline{U} \cap \overline{V})) &= (U \cup V)^\circ \cap (X \backslash \overline{U} \cup X \backslash \overline{V}) \\
&= (U \cup V)^\circ \cap ((X \backslash U)^\circ \cup (X \backslash V)^\circ) \\
&= ((U \cup V)^\circ \cap (X \backslash U)^\circ) \cup ((U \cup V)^\circ \cap (X \backslash V)^\circ) \\
&= ((U \cup V) \cap X \backslash U)^\circ \cup ((U \cup V) \cap X \backslash V)^\circ \\
&= (U \backslash U \cup V \backslash U)^\circ \cup (U \backslash V \cup V \backslash V)^\circ \\
&= (V \backslash U)^\circ \cup (U \backslash V)^\circ \\
&\subset V^\circ \cup U^\circ = U^\circ \cup V^\circ.
\end{aligned}
$$

Thus if $(b)$ holds, so does $(a)$.  □

**Lemma 2.** *If $U$ and $V$ are arbitrary subsets of $X$, then the statements $(a)$ and $(b)$ are equivalent:*

$(a)$  $\overline{U} \cap \overline{V} = \overline{U \cap V}$;
$(b)$  $((X \backslash U) \cup (X \backslash V))^\circ = (X \backslash U)^\circ \cup (X \backslash V)^\circ$.

**Proof.** By substituting $X \backslash V$ for $V$ in $(a)$ and $(b)$ of Theorem 5, we can prove the equivalence of both conditions $(a)$ and $(b)$.  □

Now we introduce a new necessary and sufficient condition different from the conditions $(b)$ and $(c)$ of Theorem 6, which may be considered as a duality of condition $(b)$ in Theorem 11.

**Theorem 12.** *If $U$ and $V$ are arbitrary subsets of $X$, then the statements $(a)$ and $(b)$ are equivalent:*

$(a)$ $\overline{U} \cap \overline{V} = \overline{U \cap V}$;
$(b)$ $\partial U \cap \partial V \subset \partial(U \cap V)$.

**Proof.** It follows from Lemma 2 that $\overline{U} \cap \overline{V} = \overline{U \cap V}$ is equivalent to $(b)$ of Lemma 2, that is,

$$((X \backslash U) \cup (X \backslash V))^{\circ} = (X \backslash U)^{\circ} \cup (X \backslash V)^{\circ}.$$

And by substituting $X \backslash U$ for $U$ and $X \backslash V$ for $V$ in Theorem 11 and then by using Theorem 3 $(b)$, the last equality is equivalent to

$$\partial(X \backslash U) \cap \partial(X \backslash V) \subset \partial((X \backslash U) \cup (X \backslash V)),$$

i.e., $\partial U \cap \partial V \subset \partial(X \backslash (U \cap V)) = \partial(U \cap V)$. □

**Remark 1.** *In general, one of the conditions $\overline{U} \cap \overline{V} = \overline{U \cap V}$ and $U^{\circ} \cup V^{\circ} = (U \cup V)^{\circ}$ does not imply the other. For example, if $U = [-1, 0]$ and $V = [0, 1]$, then the first condition holds but the second condition fails. If $U = (-\infty, 0)$ and $V = (0, \infty)$, then the second condition holds but the first one fails.*

**Open problem**. *Assume that $U$ and $V$ are arbitrary subsets of $X$ with the properties:*

$(a)$ *$U$ is an open subset of $X$;*
$(b)$ *$V$ is a closed subset of $X$;*
$(c)$ *$U$ and $V$ are disjoint.*

*Under what conditions, is $U \cup V$ an open subset of $X$ or a closed subset of $X$?*

## 7. Conclusions

The inclusion (1) or (2) holds true for 'strict inclusion' when $U$ and $V$ are some particular subsets of a topological space $X$. It seems important in many practical applications to know the condition that equal sign in the inclusion (1) or (2) holds. It is unfortunate that we were able to find a limited number of literatures related to this topic. One of the important goals of our paper is to find the necessary and sufficient conditions to solve this problem. Furthermore, the authors have proved the relations $U^{\circ} \backslash \overline{V} = (U \backslash V)^{\circ}$ and $\overline{U} \backslash V^{\circ} = \overline{U \backslash V}$ when $U$ and $V$ are subsets of $X$ under condition $(b)$, $(c)$, or $(d)$ of Theorem 5. These equalities are expected to be of great practical use in studying subjects related to general topology; for example, they can be used to demonstrate the openness of intersection of two subsets (refer to Theorem 7).

**Author Contributions:** All authors contributed equally to the writing of this paper. All authors read and approved the final manuscript. Writing original draft, S.-M.J. and D.N.; Writing review & editing, S.-M.J. and D.N.

**Funding:** This research was supported by Basic Science Research Program through the National Research Foundation of Korea (NRF) funded by the Ministry of Education (No. 2016R1D1A1B03931061).

**Acknowledgments:** This work was supported by 2019 Hongik University Research Fund.

**Conflicts of Interest:** The authors declare that there is no conflict of interest regarding the publication of this article.

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
