# Peer review of "Some Properties of Interior and Closure in General Topology"

_mathematics, doi:10.3390/math7070624_

Round 1
Reviewer 1 Report
Authors discuss about the necessary and sufficient conditions where the intersection of an open set and a closed set becomes either an open set or a closed set. As their dualities, they further introduce the necessary and sufficient conditions that the union of a closed set and an open set becomes either a closed set or an open set.
This seems to be a very challenging topic and area but there are some issues.
Authors fail to prove their findings as they cannot prove the necessity of this work and the application in different fields.
Despite the fact that the steps and the mathematical types are detailed in a thorough way showing readers the potential of this area, the research appears to be incomplete in terms of thorough explanations. More to this point, I think that the Introduction section should ideally be enriched so as to further explain to users who are not very familiar with this field, the purpose of appreciating the precise contribution made by this paper.
Furthermore, the bibliography is vital to be added and further explained and analyzed.
Authors cannot state the differences among their work and other works in the field.
What is more, I think that the paper lacks the interpretation in terms of equations and mathematical equations. That is why to use the corresponding equations?
There is also nothing about the experimental evaluation.
Moreover, readers cannot understand the applicability of the proposed method, and maybe authors should add something related to this.
In addition, authors could add further topics for future studies – maybe in a new paragraph so as to further enrich their contribution.
Author Response
I am really grateful to the reviewer for carefully pointing out what I need to improve on the content of my paper. The revised matters are summarized below according to the items indicated by the reviewer.
1. The reviewer comments: Authors fail to prove their findings as they cannot prove the necessity of this work and the application in different fields.
Answer: In pure mathematics, we do not study only the theory necessary for any application. In this paper, we did not consider any usability. The completeness of the theory is satisfactory enough. I am very sorry that I cannot describe the applicability of this paper properly.
2. The reviewer comments: Despite the fact that the steps and the mathematical types are detailed in a thorough way showing readers the potential of this area, the research appears to be incomplete in terms of thorough explanations. More to this point, I think that the Introduction section should ideally be enriched so as to further explain to users who are not very familiar with this field, the purpose of appreciating the precise contribution made by this paper.
Answer: The introduction was described in more detail as the first reviewer suggested (see the Introduction of the revised manuscript).
3. The reviewer comments: Furthermore, the bibliography is vital to be added and further explained and analyzed.
Answer: It is very difficult to find publications related to the subject of this paper. I am very sorry that I cannot find publications related to the subject of this paper.
4. The reviewer comments: Authors cannot state the differences among their work and other works in the field.
Answer: This present paper has been written based on the first author's 2016 paper [1], and the present paper has been completed with many enhancements and extensions of the previous paper [1] as we already mentioned in the discussion section. We could not find any other paper on this subject. Therefore, we are very sorry that we can not compare this article with the contents of other papers.
5. The reviewer comments: What is more, I think that the paper lacks the interpretation in terms of equations and mathematical equations. That is why to use the corresponding equations?
Answer: Theorems 4, 5, and 6, which include the equalities, are very important in the theoretical aspects. Although it is not clear at this point in what areas these equalities can be used, these equalities are very interesting from a theoretical point of view. (See the first paragraph of the revised manuscript).
6. The reviewer comments: There is also nothing about the experimental evaluation.
Answer: There is no experimental evaluation because it is not an experimental paper.
7. The reviewer comments: Moreover, readers cannot understand the applicability of the proposed method, and maybe authors should add something related to this.
Answer: Theorems 4, 5, and 6, which include the equalities, are very important in the theoretical aspects. Although it is not clear at this point in what areas these equalities can be used, these equalities are very interesting from a theoretical point of view. (See the first paragraph of the revised manuscript).
8. The reviewer comments: In addition, authors could add further topics for future studies - maybe in a new paragraph so as to further enrich their contribution.
Answer: Pease see the Open Problem described at the end of the paper, although it is not abundant.
Reviewer 2 Report
The paper may have scientific value but it lacks originality. And it is not clear whether the aimed sets are relevant in practice, as the results lack completely without supporting the claims.
This must be significantly improved.
Also, the derivation seems sequential. If one condition does not hold, then the entire subset of lemma and subsequent proofs become obsolete. This must be clarified as it limits the application of the theory.
Reference list seem outdated and very poor.
Author Response
I am really grateful to the reviewer for carefully pointing out what I need to improve on the content of my paper. The revised matters are summarized below according to the items indicated by the reviewer.
1. The reviewer comments: The paper may have scientific value but it lacks originality. And it is not clear whether the aimed sets are relevant in practice, as the results lack completely without supporting the claims. This must be significantly improved.
Answer: I agree with the opinion of the reviewer to some extent. However, this paper treats pure theory and does not aim at applicability or usability. As mentioned earlier in Theorems 4 and 6, all convex sets in $\mathbb{R}^3$ satisfy the given conditions.
2. The reviewer comments: Also, the derivation seems sequential. If one condition does not hold, then the entire subset of lemma and subsequent proofs become obsolete. This must be clarified as it limits the application of the theory.
Answer: Most of the conditions introduced in this paper are necessary and sufficient conditions and tend to be as the reviewer has pointed out. Thank you for your comments and advice.
3. The reviewer comments: Reference list seem outdated and very poor.
Answer: It is very difficult to find publications related to the subject of this paper. I am very sorry that I cannot find publications related to the subject of this paper.
Reviewer 3 Report
Broader introduction.
Broader conclusions with future works.
More references.
Author Response
I am really grateful to the reviewer for carefully pointing out what I need to improve on the content of my paper. The revised matters are summarized below according to the items indicated by the reviewer.
1. The reviewer comments: Broader introduction.
Answer: The introduction was described in more detail as the reviewer suggested (see the introduction of the revised manuscript).
2. The reviewer comments: Broader conclusions with future works.
Answer: Pease see the Open Problem described at the end of the paper, although it is not abundant.
3. The reviewer comments: More references.
Answer: It is very difficult to find publications related to the subject of this paper. I am very sorry that I cannot find publications related to the subject of this paper.
Round 2
Reviewer 1 Report
Authors have answered my issues. So I vote for acceptance!
Author Response
I thank to the reviewer who evaluated my paper favorably.
Reviewer 2 Report
I agree with the revised version.
Author Response

(The authors gave the same response as above.)
